# Protein Binding in Translational Antimicrobial Development-Focus on Interspecies Differences

**DOI:** 10.3390/antibiotics11070923

**Published:** 2022-07-08

**Authors:** Hifza Ahmed, Felix Bergmann, Markus Zeitlinger

**Affiliations:** 1Department of Clinical Pharmacology, Medical University of Vienna, 1090 Vienna, Austria; hifza.ahmed@meduniwien.ac.at (H.A.); felix.bergmann@meduniwien.ac.at (F.B.); 2Department of Plastic, Reconstructive and Aesthetic Surgery, Medical University of Vienna, 1090 Vienna, Austria

**Keywords:** plasma protein binding, fraction unbound, pharmacokinetics, antibiotics, interspecies differences, equilibrium dialysis, ultracentrifugation

## Abstract

Background/Introduction: Plasma protein binding (PPB) continues to be a key aspect of antibiotic development and clinical use. PPB is essential to understand several properties of drug candidates, including antimicrobial activity, drug-drug interaction, drug clearance, volume of distribution, and therapeutic index. Focus areas of the review: In this review, we discuss the basics of PPB, including the main drug binding proteins i.e., Albumin and α-1-acid glycoprotein (AAG). Furthermore, we present the effects of PPB on the antimicrobial activity of antibiotics and the current role of PPB in in vitro pharmacodynamic (PD) models of antibiotics. Moreover, the effect of PPB on the PK/PD of antibiotics has been discussed in this review. A key aspect of this paper is a concise evaluation of PPB between animal species (dog, rat, mouse, rabbit and monkey) and humans. Our statistical analysis of the data available in the literature suggests a significant difference between antibiotic binding in humans and that of dogs or mice, with the majority of measurements from the pre-clinical species falling within five-fold of the human plasma value. Conversely, no significant difference in binding was found between humans and rats, rabbits, or monkeys. This information may be helpful for drug researchers to select the most relevant animal species in which the metabolism of a compound can be studied for extrapolating the results to humans. Furthermore, state-of-the-art methods for determining PPB such as equilibrium dialysis, ultracentrifugation, microdialysis, gel filtration, chromatographic methods and fluorescence spectroscopy are highlighted with their advantages and disadvantages.

## 1. Introduction

Plasma protein binding (PPB) is essential for predicting and understanding several important properties of novel antibiotics, including volume of distribution, clearance, therapeutic index and drug-drug interaction [1,2,3]. Plasma is composed of several proteins that act as transporters of exogenous and endogenous molecules throughout the circulatory system and, to a lesser extent, in the interstitial space of tissue. Among these proteins, albumin and α-1-acid glycoprotein (AAG) are responsible for most of the small molecular weight drug molecule binding, whereas globulins and lipoproteins are responsible to a lesser extent [4,5]. Figure 1 shows the interaction of drug molecules with plasma protein albumin. Together, they are believed to have a major impact on the pharmacokinetic profile of exogenous compounds, but also play a role as diagnostic markers, highlighting their potential for clinical applications [6]. Human serum albumin (HSA) is the most abundant protein in human plasma and contributes to the maintenance of pH and osmotic pressure in the bloodstream at concentrations of 500 to 700 μM [5]. HSA carries peptides, drugs, fatty acids, bilirubin and other endogenous compounds [7]. The formation of non-covalent complexes of small molecular weight molecules with HSA may prevent susceptibility to excretory pathways, including renal glomerular filtration and enzymatic reactions in the liver and bloodstream [8]. AAG, a minor acute-phase component of plasma, is primarily synthesized by hepatocytes and functions as a carrier protein mainly for basic and neutral lipophilic endogenous compounds [9].

The pharmacological effect of a certain drug is dependent on both its pharmacokinetic (PK) and pharmacodynamic (PD) properties. The effects of PPB on the PK of antibiotics have been studied extensively [10], but whether PPB also affects PD and antimicrobial activity (by reducing the non-protein-bound fraction) continues to be debated [11]. This may be due to a lack of standardized in vitro PD models to quantify the impact of PPB of antimicrobials [12]. These divergent results may be due to the use of different experimental media with different protein amounts and types [13,14] or the heterogeneous methods (time-kill curves, MIC assays, dynamic models) used to study the effects of PPB in vitro. A prerequisite for all models is the precise measurement of the fraction of free antibiotics in order to correctly evaluate the impact of PPB on the PD of an antibiotic.

Here we provide an overview of the effects of PPB on the antimicrobial activity of antibiotics and the role of PPB in in vitro PD models of antibiotics. By comparing human PPB data for a variety of drugs with PPB data from different animal species, we gain insight into the potential of extrapolating PPB and PK/PD data from animals to humans.

## 2. Protein Binding: General Principles

Several proteins are involved in the protein binding of antibiotics including plasma proteins, extracellular tissue proteins and/or intracellular tissue proteins. In vivo, drug molecules are either bound to plasma or tissue proteins, or in an unbound/free state. Moreover, PPB properties greatly vary between drugs [15,16]. Since bound drugs are too large to penetrate most biological membranes, only unbound fractions can enter the target tissue and exert the expected pharmacological effect. Hence, PPB significantly influences the PD of drugs. Acidic drugs, including β-lactam antibiotics, fluoroquinolones and nonsteroidal anti-inflammatory drugs (NSAIDs), primarily bind to albumin [7,17]. In contrast, alkaline drugs, such as verapamil, propranolol and lidocaine, bind to the acute phase protein AAG. In addition, lipophilic molecules (i.e., steroid hormones) and some acidic drugs (i.e., phenobarbital) also bind to AAG [18,19]. Other proteins such as corticosteroid binding globulin are essential for binding some specific drugs including glucocorticoids, but do not play an essential role in general drug-protein binding [20]. The PPB properties of drugs are further affected by inflammation. Albumin concentrations decrease during inflammation, whereas AAG levels increase [21]. Finally, protein binding processes are influenced by several factors such as pH, temperature and concentration of the drug, which must be considered when investigating PPB properties in vitro.

## 3. Effect of Plasma Protein Binding on Antibiotics

### 3.1. Pharmacokinetics

Only the free/unbound fraction of small molecules diffuses into the extravascular space where it exerts pharmacological activity and may cause side effects [22,23,24]. Therefore, PPB may lead to lower concentrations of free antibiotics in the target tissue, which could reduce antimicrobial activity. This effect is enhanced for certain antibiotics with more pronounced PPB [23]. On the other hand, if poorly water-soluble drugs bind to serum proteins, their tissue distribution may improve due to better solubilization. Similarly, PPB may also enhance antimicrobial efficacy, with plasma proteins acting as drug reservoirs, resulting in prolonged duration of drug concentrations above the bacterial minimum inhibitory concentration (MIC) [23,25,26]. However, the concentrations of unbound drugs may never reach therapeutic levels when their binding to plasma proteins is too high or irreversible. Therefore, the binding properties of drugs must be determined prior to the development process [27]. Some antibiotics exhibit variation in their PPB that may alter certain PK parameters, but rarely impact clinical activity [20]. Changes in PPB due to drug-drug interactions can rapidly affect unbound fractions of drugs and be a source of adverse drug reactions [20]. In summary, although the literature usually describes the total concentration of the drug in plasma, information on the free/unbound fraction is of critical importance in evaluating the antimicrobial activity of a drug [2].

### 3.2. Pharmacodynamics

The effect of PPB on PD of antibiotics can be assessed in various settings. For example, studies assessing the impact of PPB on the efficacy of an antibiotic against a particular pathogen might reveal variations in the corresponding MIC [23]. Although both bacterial growth and killing are dynamic processes, the MIC is a sometimes-inconsistent threshold value that only partially reflects the activity of an antibiotic [28,29]. In contrast, growth estimation and antibiotic-enhanced time–kill analysis provides a better estimate of antimicrobial activity than MIC [29]. Therefore, time–kill curves are recommended as the most reliable experimental method to evaluate the effects of PPB on antimicrobial activity [24,30]. Methodologically, bacterial growth media are spiked with human serum or protein supplements to account for PPB in these experiments. The actual percentage of human serum must be less than 50%, as serum alone may inhibit bacterial growth [23,25,26], but the maximum serum concentration that can be used should be assessed for the particular bacteria [24,31,32,33,34]. Table 1 provides an overview of the studies that assessed the effect of plasma protein binding on antimicrobial activity.

## 4. Methodologies for Determining PPB

Equilibrium dialysis, ultracentrifugation, microdialysis and ultrafiltration are the most common methods for separating bound and unbound fractions of a drug [53,54]. Table 2 provides a description of these methods, including their advantages and disadvantages.

Equilibrium dialysis is considered the reference method [53,54,55,56], which consists of two dialysis cells separated by a semipermeable dialysis membrane [57,58] (Figure 2). Each compartment is filled with either buffer or plasma. Subsequently, unbound drugs disseminate from the plasma samples into the protein-free buffer until equilibrium is attained. The free/unbound fraction of the drug is then directly calculated from the buffer solution [55].

Another common method for quantifying the free concentration of a drug in plasma is ultrafiltration [57,58]. An ultrafiltration (UF) system consists of two chambers that are divided by a semi-permeable filter, which has different molecular weight capacities for protein filtration. UF is based on the principle of separation of small volumes of protein-free phase, by exerting a centrifugal force to a solution with both proteins and the substance of interest that is present in the upper compartment of a special UF device [57,58]. After separation by centrifugation, the free drug concentration is determined from the protein-free ultrafiltrate located in the lower compartment of the UF device.

In vivo microdialysis allows estimation of unbound drug fractions in the interstitial fluid of various tissues, and quantification of PPB in blood [59]. Microdialysis is based on the principle of diffusion of molecules along their concentration gradient between two compartments [60]. To determine PPB, a microdialysis probe with a semi-permeable membrane at its tip is inserted into a blood vessel [58] (Figure 3). Afterwards, a dialysate buffer is flushed through the probe and the unbound fraction of a drug in plasma diffuses through the semi-permeable membrane. Larger molecules, including proteins such as albumin, cannot pass through the membrane. The maximum size of proteins that can diffuse depends on the molecular weight limit of the semipermeable membrane [61]. The collected microdialysate samples can then be analyzed [58,62].

Ultracentrifugation is commonly used to estimate the free fraction of lipophilic compounds, by dividing human plasma into three layers by ultracentrifugation, which are then analyzed individually [73] (Figure 4).

In gel filtration (originally called the “zonal” method), after passing through a dextran gel column, the drug-protein solution is divided into two sections, one containing the free drug and the other containing the protein and protein-bound drug [74]. However, this technique may underestimate PPB because the bound drug separates from the drug-protein complex during passage through the column [74].

Several chromatographic methods are available for the separation of different proteins, including high-performance liquid chromatography (HPLC) and gel permeation chromatography (GPC). Chromatographic separation through an HSA-immobilized column allows relative ranking by percent binding. Moreover, equilibrium gel filtration (EGF) with liquid chromatography-tandem mass spectrometry detection has been suggested to be used earlier in the drug discovery process to measure the free concentrations of highly bound drugs without the need for radiolabeled compounds [75,76]. EGF provides an extra option for compounds that are unable to achieve equilibrium, that bind with high affinity to saturable proteins in diluted plasma, or that are unstable in plasma [75,76]. EGF consists of equilibrating a dextran column with a small molecule of interest and then introducing to the column a buffer solution containing a macromolecule capable of binding to the small molecule. Fluorescence-based spectroscopy is another method for determining the binding of small molecules to human proteins. This technique relies on small fluorescent molecules with unique spectroscopic and binding properties. Fluorescence intensity differences between the bound and unbound protein can then be detected assess PPB [77].

## 5. Assessment of Protein Binding of Antibiotics in the Serum and Plasma across Species

For preclinical findings to be translated into clinical applications, the differences in physiological processes between animals and humans are important and must first be understood [41,78,79]. The quality and accuracy of preclinical estimates depend on the similarities in physiology, drug metabolism, distribution, and absorption between animals and humans [80,81]. Though there have been great advancements in pharmacokinetics in recent years, it still remains a challenge to predict all of the pharmacokinetic parameters of a drug in humans from animal studies. Nonetheless, it is possible to make reasonably good predictions under some well-defined conditions. For instance, the intrinsic absorption of a certain drug through the wall of the gastrointestinal tract might be comparable across species, as the nature of the biomembrane of epithelial cells is similar in mammals [82]. Moreover, the absorption process is an interaction between the drug and the biomembrane that also plays its role. However, there are other factors, including pH-dependent solubility and first-pass metabolism that affect absorption and can lead to differences in absorption in different animal species. One example of relatively successful prediction is the renal excretion of drugs in humans using the glomerular filtration rate ratio between humans and animals. Similarly, if the drug is primarily excreted by the liver and the rate of excretion is limited by blood flow in the liver, the clearance of the drug in humans can be predicted by blood flow in the liver. Nevertheless, biochemical parameters, such as protein binding and drug metabolism, are less predictable, and vary considerably among species [80,81,82]. Rodents are often selected for such studies because they have a short lifespan and therefore a large number of animals can be bred quickly, which in turn allows multiple studies to be conducted. [83]. Body weight and height are usually considered important co-variables for the determination of key PK parameters. [84]. Therefore, small animals can be expected to excrete drugs faster than humans when compared on a weight-normalized basis. Other physiological parameters such as body temperature and serum albumin concentration are comparable in different animals and independent of animal size [85].

Table 3 provides an interspecies comparison of protein binding in serum and plasma in humans and in various animals. We performed a Wilcoxon matched-pairs signed rank test between human and animal PPB data and found that there was a significant difference between antibiotic binding in humans and dogs or mice. In contrast, no significant difference in binding was found between humans and rats, rabbits, or monkeys (Table 3). However, in monkeys, only four pairs were included in the analysis, thus limiting the generalizability of our findings. We further calculated the ratio of antibiotic PPB in humans and various animals (Table 4). The greatest difference was observed in dogs (ratio human vs. dog: 1.96). In general, there is a reasonable positive correlation between the PPBs detected for drug molecules in human plasma and those of rats, rabbits, and monkeys, although slightly stronger binding to human plasma proteins was observed compared with those of preclinical species (Figure 5). Therefore, for screening purposes, data collected in rats may provide a suitable initial surrogate for human protein binding. We believe that PPB should always be measured in species of interest for further drug development, such as for studying pharmacokinetics in preclinical settings, for understanding pharmacodynamics in disease models, or for making dose-human predictions.

## 6. Conclusions

PPB has received little attention in the discovery and development of antimicrobial agents in recent decades. Equilibrium dialysis and microdialysis are common methods used for protein binding studies. Designing studies to investigate PPB from a PK or PD perspective and generating adequate PPB data in the preclinical setting must be carefully planned, especially when a new drug project enters the investigational phase.

The present review provides information on whether animal models are adequate predictors of human PPB by analyzing data from the literature. Rats, rabbits, and monkeys exhibited similar PPB characteristics to humans, whereas significant differences were found in dogs and mice compared with humans. Depending on the drug and drug delivery platform, species-specific physiological differences may result in incorrect extrapolation of drug PK, safety and efficacy. On the other hand, interspecies differences in protein binding as a result of PK variations could indicate potential variations in drug effects when used in human patients. Therefore, we advocate the standardization of experimental settings to study PPB and its effect on PK/PD of novel antibiotics. In addition, we recognize the relevance of interspecies differences in PPB but appreciate the potential of extrapolating from preclinical pharmacology and safety studies in different species to estimate outcomes in humans.

## Figures and Tables

**Figure 1 antibiotics-11-00923-f001:**
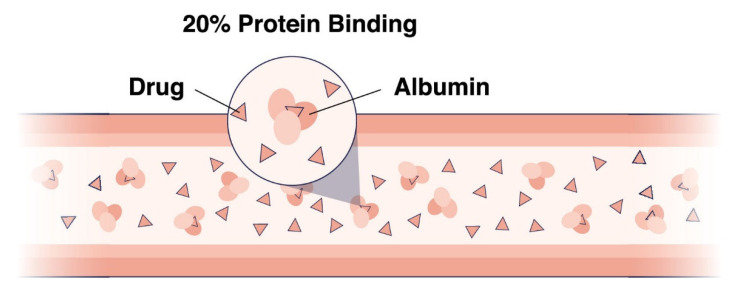
Interaction of drug molecules with plasma protein albumin.

**Figure 2 antibiotics-11-00923-f002:**
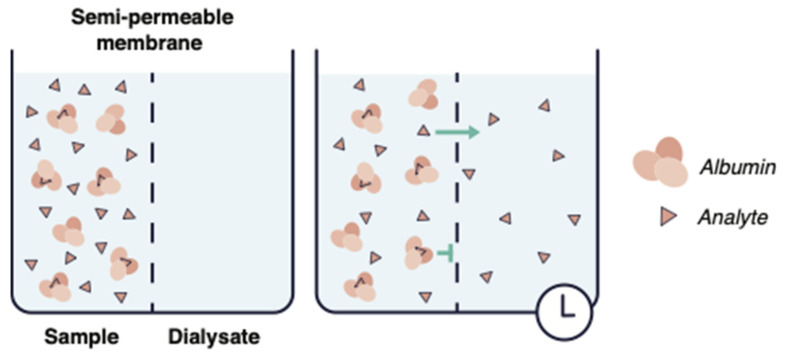
Overview and principle of equilibrium dialysis.

**Figure 3 antibiotics-11-00923-f003:**
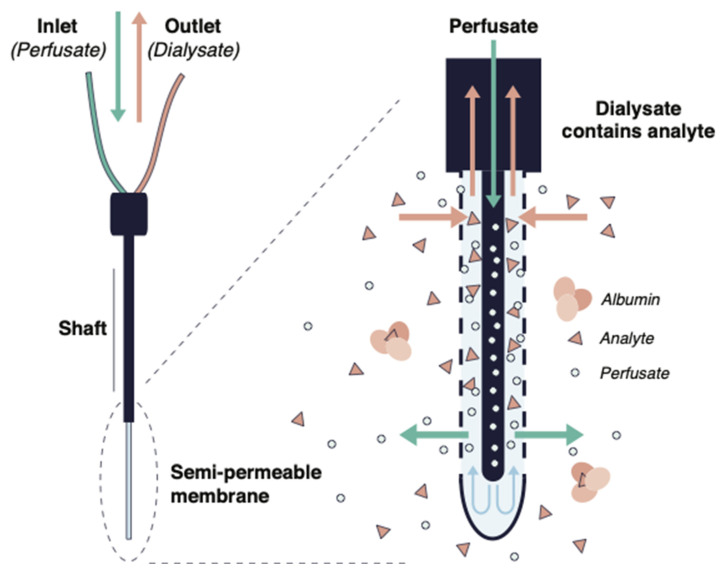
Illustration of the principle of in vitro microdialysis.

**Figure 4 antibiotics-11-00923-f004:**
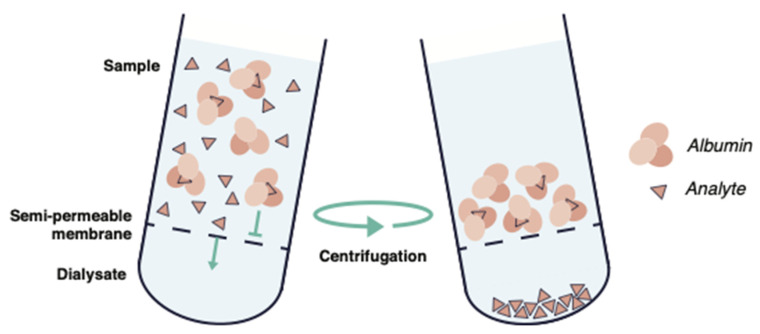
Schematic illustration of ultracentrifugation.

**Figure 5 antibiotics-11-00923-f005:**
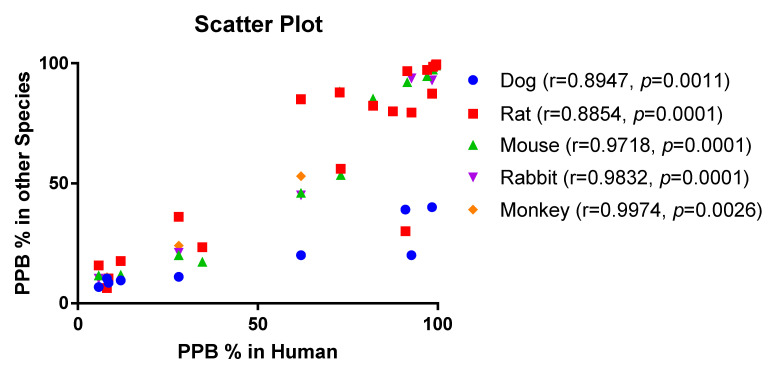
Scatter plot depicting the correlation between % plasma protein binding in humans and other species. r is the Pearson correlation coefficient.

**Table 1 antibiotics-11-00923-t001:** Examples for studies that assessed the effect of plasma protein binding on antimicrobial activity.

Antibiotic Class	Antibiotic	Plasma Protein Binding % (Drug Concentration)	Protein Preparation	Medium	Techniques for Protein Quantification	Susceptibility Test/s	Pathogens	References
Cephalosporin	Cefepime Ceftazidime Cefotaxime Ceftriaxone	19% 21% 38% 84–96% (70–300 µg/mL)	HA (4 g/dL)	SMHB	NA	Time–kill curves, MIC, MBEC	*E. aerogenes* *K. pneumoniae* *S. aureus*	[35,36,37,38,39]
	Ceftriaxone	84–96% (70–300 µg/mL)	20% Human serum	BHI	NA	Time–kill curves	*S. aureus* *P. aeruginosa*	[38,40]
	Ceftriaxone Ertapenem	HA 76.8 ± 11.0%; BSA, 20.2 ± 8.3%; HSA 56.9 ± 16.6%, HA 73.8 ± 11.6%; BSA 12.4 ± 4.8%; HSA 17.8 ± 11.5%	BSA and HSA (40 g/L)	MHB, THB + 5% CO_2_	in vitro microdialysis	MIC, time–kill curve	*E. coli* *S. pneumoniae*	[41]
	Cefditoren	88%	90% HS, 4 g/dL HA	MHB	NA	Time–kill curve	*S. pneumoniae*	[42,43]
	Cefotaxime	10–40% (0.5–32 µg/mL)	90% Pooled human CSF	CAMHB	NA	MIC	*E. coli*	[38,44]
	Ceftriaxone, Cefoperazone, Moxalactam, Ceftizoxime	92.2% 89.7% 63.8% 29.4%	HA (0, 2.5, or 5% solution), heat-inactivated HA (0, 25, 50, or 95%) (95% Human Serum)	MHB	Equilibrium dialysis	MIC, MBEC	*S. aureus* *E. coli* *P. aeruginosa* *K. pneumoniae*	[26]
Penicillin	Ampicillin, Oxacillin	20% 60–94%	40 g/L human albumin	MHB	NA	Time–kill curve	*S. aureus*	[24,45]
Carbapenem	Ertapenem	85–95% (50–300 µg/mL)	50% pooled human plasma	TSB. CAMHB	Surface Plasmon Resonance (SPR) assay	MIC	*S. aureus*	[46]
Fluoroquinolones	Ciprofloxacin	20–40% (2 µg/mL)	90% pooled human CSF	CAMHB	NA	MIC	*E. coli*	[17,44]
	Moxifloxacin, Trovafloxacin	38% (0.2–5 µg/mL) 77% (0.2–5 µg/mL)	HS (20%, 70%, 100%), HA (4%, 8%, 12%, 16%)	MHB	Ultrafiltration	MIC	*S. aureus* *P. aeruginosa*	[14]
	Ciprofloxacin, Moxifloxacin	20–40% (2 µg/mL) 26–30% (1–5 µg/mL)	rat polyvinyl sponge model	BHI, RPMI 1640 (for cells)	NA	Viable cell count	*P. aeruginosa* *S. pneumoniae*	[14,17,47]
	Moxifloxacin Ciprofloxacin Trovafloxacin	20–40% (2 µg/mL) 26–30% (1–5 µg/mL) 77% (0.2–5 µg/mL)	HA 10%, 30%, 50%	MHB	NA	MIC	*S. pneumoniae* *S. aureus* *E. coli*	[13,14,17]
Diaminopyrimidine	Iclaprim	93%	50% HP	MHB	NA	MIC	*S. aureus*	[48]
Cyclic lipopeptide	Daptomycin	91.7%	90% HS 4 g/dL HA	MHB	NA	Time–kill curves	*S. pneumoniae* *E. faecium*	[34,49]
	Daptomycin	91.7%	50% HS	MHB	NA	MIC, time–kill curve	*S. aureus* *E. faecium*	[34,50]
	Daptomycin	91.7%	4 g/dL HA	CAMHB	NA	Time–kill curves	MRSA	[34,51]
Glycopeptide	Vancomycin	36.9%	4 g/dL HA	Cation-adjusted MHB	NA	Time–kill curves	MRSA	[51,52]

NA: Not Available; BHI: Brain Heart Infusion; MRSA: Methicillin-resistant Staphylococcus aureus; HA: Human albumin; BSA: Bovine Serum Albumin; HSA: Human Serum Albumin; MHB: Mueller–Hinton Broth; THB: Todd Hewitt broth; MIC: Minimal Inhibitory Concentration; TSB: Tryptone Soy Broth; CAMHB: Cation-Adjusted Mueller–Hinton Broth; HPLC: High-performance liquid chromatography; MBEC: Minimal Biofilm Eradication Concentration; RPMI: Roswell Park Memorial Institute (Growth Media); HP: Haptoglo.

**Table 2 antibiotics-11-00923-t002:** Comparison of methods for plasma protein binding (PPB).

Technique/Method	Principle	Advantages	Drawbacks/Issues	References
Ultrafiltration	Plasma water and unbound drug is forced through a semipermeable filter, retaining protein–drug complexes	Technically simple to perform Requires a small amount of sample Available commercially via kits Quick Does not require the use of unphysiological buffer Inexpensive Suitable for unstable drugs Most commonly used	Ultrafiltrate volumes should be ≤40% of the initial plasma sample because of changes in protein concentration Leakage of membrane may happen Membrane adsorption of drugs Partly uncontrolled temperature	[57,58,63,64,65,66]
Equilibrium dialysis	Separated by a semipermeable membrane, unbound drug diffuses from plasma into protein-free buffer, until equilibrium is reached	Gold standard method Simple Reliable results Temperature controlled and thermodynamically sound	Requires the use of unphysiological buffer Non-specific binding to dialysis device and membrane Volume shifts and pH changes Membrane adsorption Requires time to reach equilibrium Drug stability concerns	[56,58,60,65,66,67,68,69]
Microdialysis	Dialysate buffer is driven through an embedded probe, having a microdialysis membrane. Unbound drug disperses from blood into dialysate	In vivo protein binding determination Volume shifts or dilution effects are lacking May also be used to find free tissue levels	Probe implantation and fixation Altering drug concentration over a period of time Membrane adsorption Needs equilibration in vivo For measurement of PPB of total antibiotic concentration in vivo is needed	[41,56,58,61,62]
Ultracentrifugation	Dissociation of protein and low-molecular-weight components occurs only by gravitation (centrifugation)	Simple Lack of membrane adsorption, dilution, volume shifts, drug–protein leakage Does not require use of unphysiological values Buffer	Long time span for sample preparation (overnight) Concentration gradient from bottom to top can result in false high binding Less suitable for high molecular weight substances Sedimentation, back diffusion Expensive equipment	[55,56,65,67,70]
Gel filtration	The free and not the total drug concentration is the independent variable.	Robust and high resolution technique Desalination	Costly Time-consuming	[71]
Chromatographic methods	Chromatographic methods include a range of techniques, based on separation of substances (including the bound and unbound fraction of an antibiotic) on the basis of different physical or chemical properties such as molecular size, charge, affinity etc.	Relatively low sample consumption Accurate methods Absence of membrane adsorption, dilution, volume shifts, drug–protein leakage	Expensive and elaborate Poorly sensitive for drugs with low affinity binding	[72]
Fluorescence spectroscopy	Higher energy photons are used to excite a sample, which then emit lower energy photons. The change in fluorescence at changing ligand/protein concentrations is used to calculate the concentration of bound drug.	Enables direct determination of bound drug concentrations	Poorly sensitive for drugs with low-affinity binding Elaborate technique	[56]

**Table 3 antibiotics-11-00923-t003:** Interspecies comparison of protein binding in serum and plasma. Wilcoxon matched-pairs signed rank test was performed between antibiotics percentage unbound in human and other species. *p*-Value < 0.05 is considered as significant.

For Serum (% Bound)								
Antibiotic	Concentration (µg/mL)	Human	Dog	Rat	Mouse	Rabbit	Monkey	References
Cefotetan ^#^		91.0	39	30				[86]
Cefpirmide ^#^	15	98.4	40	87.4		92.9		[87]
Ceftriaxone ^#^	100	92.7	20	79.5		93.7		[39]
Cefpirome	30	5.8	6.8	15.8	11.6	10.3		[88]
Ceftazidime	30	11.9	9.5	17.6	11.9	17.6		[88]
Cefzopran	20	8.1	10.4	6.4	7.1	9.8	10.9	[89]
Cefclidin	20	8.5	8.5	10.4	9.8	7.3	8.1	[89]
Carumonam	20	28	11	36	20	21	24	[90]
Ristocetin	4–120	73.1		56.1	53.5			[91]
Oritavancin		82		82.4	85.3			[92]
Oritavancin ^#^		87.5		80				[93]
Vancomycin	4–100	34.6		23.4	17.3			[91]
Mannosylaglycone	6–170	72.8		87.9	88.3			[91]
Ardacinaglycone	6–180	91.5		96.7	92.1			[91]
Ardacin A	6–190	97		97.2	94.6			[91]
Ardacin B	6–210	98.7		98.5	97.4			[91]
Ardacin C	6–210	99.6		99.5	99.1			[91]
Pseudoaglycone	6–190	99.4		99.1	99			[91]
Aztreonam	20	62	20	85	46	45	53	[90]
Wilcoxon matched-pairs signed rank test								
Number of pairs with human values			9	19	15	8	4	
*p*-value			* (0.0195)	NS (0.8059)	* (0.0353)	NS (0.3750)	NS (0.5000)	

^#^ Plasma was used instead of serum. * *p*-value < 0.05.

**Table 4 antibiotics-11-00923-t004:** Human/animal ratio of protein binding of different antibiotics in serum and plasma. Means difference analysis was performed between antibiotics percentage bound ratio in human and other species. Mean, standard deviation and range of human/animal ratio is provided.

For Serum (% Binding Ratio)					
Antibiotic	Human/Dog	Human/Rat	Human/Mouse	Human/Rabbit	Human/Monkey
Cefotetan ^#^	2.3	3.0			
Cefpiramide ^#^	2.5	1.2		1.1	
Ceftriaxone ^#^	4.6	1.2		1.0	
Cefpirome	0.8	0.4	0.5	0.6	
Ceftazidime	1.2	0.7	1.0	0.7	
Cefzopran	0.8	1.3	1.1	0.8	0.7
Cefclidin	1.0	0.8	0.9	1.2	1.0
Carumonam	2.5	0.8	1.4	1.3	1.2
Ristocetin		1.3	1.4		
Oritavancin		1.0	1.0		
Oritavancin ^#^		1.1			
Vancomycin		1.5	2.0		
Mannosylaglycone		0.8	0.8		
Ardacinaglycone		0.9	1.0		
Ardacin A		1.0	1.0		
Ardacin B		1.0	1.0		
Ardacin C		1.0	1.0		
Pseudoaglycone		1.0	1.0		
Aztreonam	3.1	0.7	1.3	1.3	1.2
Means difference analysis					
Number of pairs with human values	9	19	15	8	4
Mean	1.96	1.09	1.09	1	1.02
Standard deviation	1.30	0.53	0.33	0.27	0.24
Range	0.8–4.6	0.4–3.0	0.5–2.0	0.6–1.3	0.7–1.2

^#^ Plasma was used instead of serum.

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
