# Peer review of "Protein Binding in Translational Antimicrobial Development-Focus on Interspecies Differences"

_antibiotics, 2022, doi:10.3390/antibiotics11070923_

Round 1

Reviewer 1 Report

The current study is quite interesting, since plasma protein binding is a very important factor that affects all drugs and their performance when administered. A comprehensive review on interspecies differences is quite relevant, since in vivo results can only be useful when able to be extrapolated to the clinical context. Moreover, the fact that no statistical significance was found in plasma protein binding between most animal species and humans, especially for the most used species in in vivo assays (mice, rats, and rabbits) is quite positive, albeit the number of analyzed articles sometimes being small (which is already acknowledged and commented by the authors). This review falls well within the scope of “Antibiotics” journal, and I recommend its publication.

Author Response

Dear Reviewer

Thank you very much for your valuable comments and recommendation of the publication.

Reviewer 2 Report

The authors et.al. described a review in detail the title of “Protein binding in translational antimicrobial development focus on interspecies differences”. The review article is very interesting to the audience and the current version of the manuscript is suitable for acceptable publication in Antibiotics.

One of the comments I am suggesting for the authors. It would be good if they included the figures describing an example of the interaction of drug molecules with Plasma protein binding.

Author Response

Dear Reviewer

Thank you very much for your valuable comments and suggestions. The suggested figure has now been added to the revised manuscript.

Reviewer 3 Report

Abstract Background/Introduction: Plasma protein binding (PPB) continues to be a key aspect of antibiotic development and clinical use. PPB is essential to understand several properties of drug candidates, including antimicrobial activity, drug-drug interaction, drug clearance, volume of distribution, and therapeutic index. Focus areas of the review: In this review, we discuss the basics of PPB, including the main drug binding proteins, i.e., Albumin and α-1-acid glycoprotein (AAG). Further, we present the effects of PPB on the antimicrobial activity of antibiotics and the current role of PPB in in vitro pharmacodynamic (PD) models of antibiotics. A key aspect of this paper is a concise evaluation of PPB between animal species (dog, rat, mouse, rabbit and monkey) and humans. Furthermore, state-of-the-art methods for determining PPB such as equilibrium dialysis, ultracentrifugation, microdialysis, gel filtration, chromatographic methods and fluorescence spectroscopy are highlighted with their advantages and disadvantages.

Comment: the abstract could be expanded to include summary of the review. Suggest to stress on information below:  Comparison of the human plasma protein binding data for a variety of drug discovery compounds indicates that compounds tend to be slightly more bound to human plasma proteins, than compared to plasma proteins from rats, dogs or mice. However, the majority of measurements from the pre-clinical species fall within 5-fold of the human plasma value, although there are some compounds that do show significantly different interspecies plasma protein binding

Protein binding: General principles :

Effect of plasma protein binding on antibiotics 

Pharmacokinetics

Pharmacodynamics

Methodologies for determining PPB

Comments: These sections are well described

Figure 1. Overview and principle of equilibrium dialysis.

Figure 2. Illustration of the principle of in vitro microdialysis.

Figure 3. Schematic illustration of ultracentrifugation

Figure 4. Scatter plot depicting the correlation between % plasma protein binding in humans and other species. r is the Pearson correlation coefficient.

Comment: Pls indicate if these figures were adopted elsewhere

Table 1. Known factors affecting plasma protein binding.-this has been described in text, so can be omitted.

Table 2. Examples for studies that assessed the effect of plasma protein binding on antimicrobial activity. –this table is good

Table 3. Comparison of methods for ?? plasma protein binding (PPB)

Table 4. Interspecies comparison of protein binding in serum and plasma. Wilcoxon matched-pairs signed rank test was performed between antibiotics percentage unbound in human and other species. P-value<0.05 is considered as significant (P-value<0.05=*, P-value<0.01=**).=this table is not easy to follow,pls simplify the info/presentation

Table 5. Human/animal ratio of protein binding of different antibiotics in serum and plasma. Means difference analysis was performed between antibiotics percentage bound ratio in human and other species. Mean, standard deviation and range of human/animal ratio is provided. Which reference was used? How these data were generated?

This is an important reference on this topic, pls justify how this review is different:

Colclough, N., Ruston, L., Wood, J. M., & MacFaul, P. A. (2014). Species differences in drug plasma protein binding. MedChemComm5(7), 963-967.

On the other hand, interspecies differences in protein binding as a result of PK variations could indicate potential variations in drug effects when used in human patients. Therefore, we advocate the standardization of experimental settings to study PPB and its effect on PK/PD of novel antibiotics. In addition, we recognize the relevance of interspecies differences in PPB but appreciate the potential of extrapolating from preclinical pharmacology and safety studies in different species to estimate outcomes in humans.

Comment: any existing guideline related to this? Is this practical to do so

Author Response

Response to Specific Comments:

Reviewer 3

Comment: the abstract could be expanded to include summary of the review. Suggest to stress on information below:  Comparison of the human plasma protein binding data for a variety of drug discovery compounds indicates that compounds tend to be slightly more bound to human plasma proteins, than compared to plasma proteins from rats, dogs or mice. However, the majority of measurements from the pre-clinical species fall within 5-fold of the human plasma value

, although there are some compounds that do show significantly different interspecies plasma protein binding.

Response: The abstract has been modified accordingly and the suggested information has been added accordingly.

Comment:

Figure 1. Overview and principle of equilibrium dialysis.

Figure 2. Illustration of the principle of in vitro microdialysis.

Figure 3. Schematic illustration of ultracentrifugation

Figure 4. Scatter plot depicting the correlation between % plasma protein binding in humans and other species. r is the Pearson correlation coefficient.

Pls indicate if these figures were adopted elsewhere.

Response: Figure 1, 2 and 3 were adapted from different studied. Those studies have been cited in the revised manuscript. Figure 4 has been constructed from the data available in the literature (in Table 4) and all the references are added in the table.

Comment: Table 1. Known factors affecting plasma protein binding.-this has been described in text, so can be omitted.

Response: Table 1 has been removed as suggested.

Comment: Table 3. Comparison of methods for ?? plasma protein binding (PPB)

Response: Yes, for PPB.

Comment: Table 4. Interspecies comparison of protein binding in serum and plasma. Wilcoxon matched-pairs signed rank test was performed between antibiotics percentage unbound in human and other species. P-value<0.05 is considered as significant (P-value<0.05=*, P-value<0.01=**).=this table is not easy to follow,pls simplify the info/presentation.

Response: Table 4 contains PPB data in serum/plasma in humans and other species. We have got this data from different studies available (references give for each study) and performed statistical analysis to check the statistical significance of the data. The data provides information if PPB in humans is similar or different from that in other species, so we think the table is really important for the paper as it is.

Comment: Table 5. Human/animal ratio of protein binding of different antibiotics in serum and plasma. Means difference analysis was performed between antibiotics percentage bound ratio in human and other species. Mean, standard deviation and range of human/animal ratio is provided. Which reference was used? How these data were generated?

Response: These data were taken from different studies and the reference of each study is provided in Table 4. The values in Table 4 have been used for the calculation of ratios.

Comment: is an important reference on this topic, pls justify how this review is different:

Colclough, N., Ruston, L., Wood, J. M., & MacFaul, P. A. (2014). Species differences in drug plasma protein binding. MedChemComm, 5(7), 963-967.

Response: The reference mentioned above is a nice study of 2014 showing species differences in drug plasma protein binding. We have tried to include recent studies as well and there are other aspects that have been in our review including methodologies for determining PPB, principles and role of PK/PD in PPB. Moreover, we have performed statistical analysis on the data available on interspecies PPB.

Comment: On the other hand, interspecies differences in protein binding as a result of PK variations could indicate potential variations in drug effects when used in human patients. Therefore, we advocate the standardization of experimental settings to study PPB and its effect on PK/PD of novel antibiotics. In addition, we recognize the relevance of interspecies differences in PPB but appreciate the potential of extrapolating from preclinical pharmacology and safety studies in different species to estimate outcomes in humans.

Comment: any existing guideline related to this? Is this practical to do so?

Response: This is debatable as there may be few recommendations but they also vary depending upon the testing conditions and that is why still a lot more needs to be done to reach to a level where we can formulate hard and fast rules and guidelines.